# Creating Inclusive Schools for Autistic Students: A Scoping Review on Elements Contributing to Strengths-Based Approaches

Jia White [1,2,*], Sarah McGarry [3], Marita Falkmer [4], Melissa Scott [3], P. John Williams [1] and Melissa H. Black [3,5]

1  School of Education, Curtin University, Perth, WA 6102, Australia; pjohn.williams@curtin.edu.au
2  Autism CRC, Brisbane, QLD 4068, Australia
3  Curtin School of Allied Health, Curtin University, Perth, WA 6102, Australia; sarah.mcgarry@curtin.edu.au (S.M.); melissa.scott@curtin.edu.au (M.S.); melissa.black@ki.se (M.H.B.)
4  CHILD, Swedish Institute for Disability Research, School of Education and Communication, Jönköping University, 553 18 Jönköping, Sweden; marita@falkmer.se
5  Center of Neurodevelopmental Disorders (KIND), Centre for Psychiatry Research, Department of Women's and Children's Health, Karolinska Institutet & Stockholm Health Care Services, Region Stockholm, 171 77 Stockholm, Sweden
*  Correspondence: jia.white@curtin.edu.au

**Abstract:** Strengths-based approaches leveraging the strengths and interests of autistic students are increasingly recognised as important to meeting their school-related needs. A scoping review exploring elements contributing to strengths-based approaches for autistic students in schools was undertaken. Eighteen articles were identified, with results conceptualised according to the Bioecological Model of Development. One personal (strengths and interests), six microsystem (specialised instructions, curriculum integration, curriculum differentiation, common interests with peers, reciprocal roles and adult involvement), three mesosystem (matching resources and activities, real-life learning experiences and benefiting all students), and three exosystem (cost-effective and timesaving, collaboration with colleagues and parents and teachers' attitude and knowledge) elements were identified. Findings highlight the interrelatedness of the elements contributing to strengths-based approaches for autistic students, which can be used to aid in the development of more inclusive school environments.

**Keywords:** autism; strength-based; bioecological model of development; inclusive school; scoping review

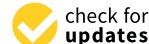



## 1. Introduction

Inclusive schooling for all students, including autistic students, has become a global effort since the Salamanca Statement and Framework for Action on Special Needs Education was published in 1994 by the United Nations Educational, Scientific and Cultural Organization (UNESCO), reaffirming the commitment to educating children, youth and adults with special educational needs [1,2]. As part of this global effort, there has been an increase in the number of autistic students in mainstream schools in recent years [3]. In Australia, there were 205,200 autistic individuals in 2018, a 25.1% increase from 2015, including 106,600 autistic young people aged 5–20 years attending schools or educational programs [4]. Many autistic students with average or above average cognitive abilities attend regular education classrooms with their neurotypical peers [3,5,6], but often face difficulties in academic learning, communication and socialising resulting in negative school experiences such as bullying and anxiety [4,7,8].

Given that inclusive education has demonstrated some benefits for students on the autism spectrum as well as for neurotypical students, interventions have been implemented in an attempt to address the difficulties autistic students face in school [1–3]. However,

most of these interventions have utilised a traditional deficit-based approach, i.e., focusing on the individual and their diagnosis alone for both analysing the causes of their challenges and determining expected outcomes [4]. While deficit-based interventions have shown effectiveness in improving executive function skills [5] and communication skills to a certain degree [6], they may have negative impacts on autistic students' mental health, school experiences and outcomes. This may be due to the focus being exclusively on the weaknesses of autistic students, ignoring the positive characteristics and strengths of autistic students that allow them to flourish [4,7]. Indeed, despite these interventions, autistic students still report negative school experiences, particularly low school connectedness [8]. Moreover, autistic students have expressed a desire to be able to use what they know to help others [9], have opportunities to interact with peers through their mutual interests [10] and contribute to group activities [11]. These desires indicate the importance of exploring, applying and leveraging the strengths and interests of autistic students to improve their overall educational experiences and outcomes. A paradigm shift from a deficit-focused approach to a strengths-based approach is needed to develop inclusive school environments that unpack the strengths of autistic students, address school-related challenges and better provide for the needs of autistic students. This is particularly applicable for autistic students in high schools, as researchers have discussed that rigorous learning opportunities, connections to students' post-school pathways and supportive relationships are three equally important aspects for both successful in-school experiences and the transition to adulthood [12,13].

Strengths-based approaches, originally used in social work, are approaches based on the philosophical principles of social justice that emphasise self-determination and empowering individuals to make changes in their own lives by acknowledging "what is right" within people, drawing upon the unique strengths, interests and preferences of individuals as well as the resources available in their environment [14–16]. Preliminary research suggests that strengths-based programs and interventions have many benefits to autistic students, such as improving relationships with family members [17,18], improving socialisation [19] and enhancing academic performance and vocational outcomes [20,21]. However, there is a paucity of research on strengths-based approaches for autistic students, especially for older students in school environments.

Although what constitutes and defines a strengths-based approach are varied, there appears to be multiple individual and environmental factors as its essential components [22]. Therefore, the Bioecological Model of Development, which focuses on the dynamic interactions between the characteristics of a developing person and the different layers of the environment over time and is commonly applied in education settings, would provide a useful theoretical framework to understand the elements contributing to strengths-based approaches for autistic students [23–25] (Figure 1). The person is always at the centre, with their characteristics of demand (age, gender and health), resource (skills, knowledge and previous experiences) and force (motivation and temperament) being impacted and formed through the interactions with the environment [23–25]. The environment that the developing person interacts with has different layers. The microsystem is the immediate environment where the proximal interactions occur, for instance, classrooms, schools and families. The mesosystem represents the relationships between the different microsystems. The exosystem is the next layer of context, in which, although the developing person is not directly involved, has indirect influences on the person. The macrosystem is the broadest level of the context that includes the cultural values and beliefs, political and social policies in a society. Bronfenbrenner also specified another dimension of context, the chronosystem, which refers to time both within a person's lifetime and the historical context [23–25]. Through the lens of the Bioecological Model of Development, autism needs to be understood as a developmental process between a person and the environment rather than an inner condition that causes deficits [26].

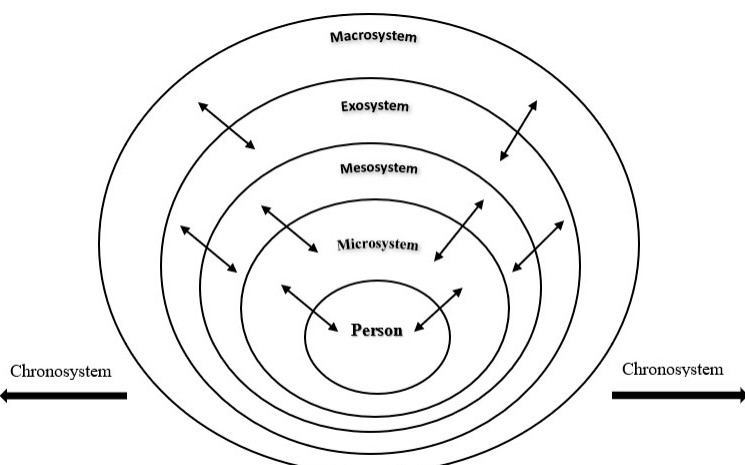

**Figure 1.** Visual illustration of the Bioecological Model of Development, adapted from Duchesne et al. [27].

This scoping review examined the literature related to strengths-based approaches for autistic students in schools, summarising and synthesising findings to inform future practices and research that promote more inclusive schooling experiences by leveraging student strengths. The primary objectives of this review were to employ the Bioecological Model of Development as a framework to explore literature relating to strengths-based approaches for autistic students in schools and to identify key elements contributing to the design and implementation of strengths-based approaches for autistic students.

## 2. Methods

A scoping review was deemed appropriate given that strengths-based approaches for autistic students in mainstream school settings are relatively new and scoping reviews assist in identifying all relevant literature and mapping broad topics areas of research using diverse research designs [28]. The scoping review process followed five stages based on the framework proposed by Arksey and O'Malley [28]: (1) identifying the research question, aims and objectives; (2) searching for relevant studies; (3) study selection; (4) charting the data; and, (5) collating, summarising and reporting the data. For the purposes of this review, strengths-based approaches are defined as approaches that acknowledge, utilise and leverage the strengths, interests and resources of autistic students to address their needs, optimise their school experiences and improve their outcomes.

### 2.1. Eligibility Criteria

Studies were included if (i) participants were students within the age group of 8 to 18 years old attending upper primary school or high school or were educators or parents of autistic students attending upper primary or high schools (ii) students had a diagnosis of autism based on the DSM-IV or DSM-5; (iii) examined the application of strengths-based approaches in school settings; and (iv) were written in English and published in peer-reviewed journals.

### 2.2. Search Strategy

The five electronic databases of ERIC, ProQuest, PsycINFO, Web of Science and Scopus were searched for relevant literature written in English and published between 1994 and June 2023. This time period was chosen because 1994 was the year when the Salamanca Statement and Framework for Action on Special Needs Education was published [29], and since then implementing inclusive schooling for all children has become a global effort [30,31]. Key search terms were grouped into four concepts including: participants, intervention, diagnosis and context as illustrated in Table 1 and were adapted to each

database. A further search of reference lists of studies was conducted through the database searches, as well as manual searching of key journals.

**Table 1.** Search terms [a].

| Participants | Intervention | Diagnosis | Context |
|---|---|---|---|
| Student *, learner *, adolescent *, teacher * | Strength * base *, strength * based practice, Strength * base * teaching practice *, strength * base * teaching method *, interest * base *, strengths-based teaching | Autism, Asperger syndrome autis *, autism spectrum disorder, asperger *, pervasive development* disorder *, autistic disorder * | School *, "mainstream school", Classroom, School-based |

[a] Terms were connected with 'OR' and between terms with 'AND'; * Search terms truncated and exploded.

### 2.3. Charting, Collating and Synthesising the Data

Data were extracted from articles into a charting framework containing descriptive entries and specific thematic information [28]. Given the diverse nature of the literature included in this review, a narrative synthesis approach was adopted to combine evidence from qualitative, quantitative, mixed-methods and research-to-practice studies to draw conclusions across the studies [32,33]. Descriptive study characteristics were extracted and organised into a data charting table by author, year, country, design, participants, context, objectives or research questions, outcome measures and outcome areas. Elements were mapped into the person and different levels of school environment based on the Bioecological Model of Development.

### 2.4. Assessment of Methodological Quality

The methodological quality of included research articles was assessed by four independent reviewers in compliance with the Standard Quality Assessment Criteria developed by Kmet et al. [34]. The checklist comprises 14 items for quantitative studies and 10 items for qualitative studies, with each item being scored two points for "criteria met", one point for "criteria partially met" and zero points for "criteria not met", adding up to an overall score of the study's methodological quality represented as a percentage. The percentage scores were then used to define the quality as strong (score of >80%), good (70–80%), adequate (50–70%) or limited (<50%) [35]. Any discrepancies between reviewers were resolved by discussion until consensus was reached.

## 3. Results

### 3.1. Search Results

As shown in Figure 2, a total of 5606 studies were identified. Following the removal of duplicates and articles that were not peer-reviewed, 5151 were screened at the title and abstract level. Eighty eligible articles were identified, and their full texts were retrieved and reviewed. After reviewing the full text, 62 further articles were excluded due to (1) the interventions or strategies not being applied to students within the age range (k = 19), (2) the settings of the interventions were not school environments (k = 11), (3) no elements of strengths-based approach or practice were included (k = 31), or (4) participants have no formal diagnosis of autism (k = 1). This resulted in 18 articles being included for review.

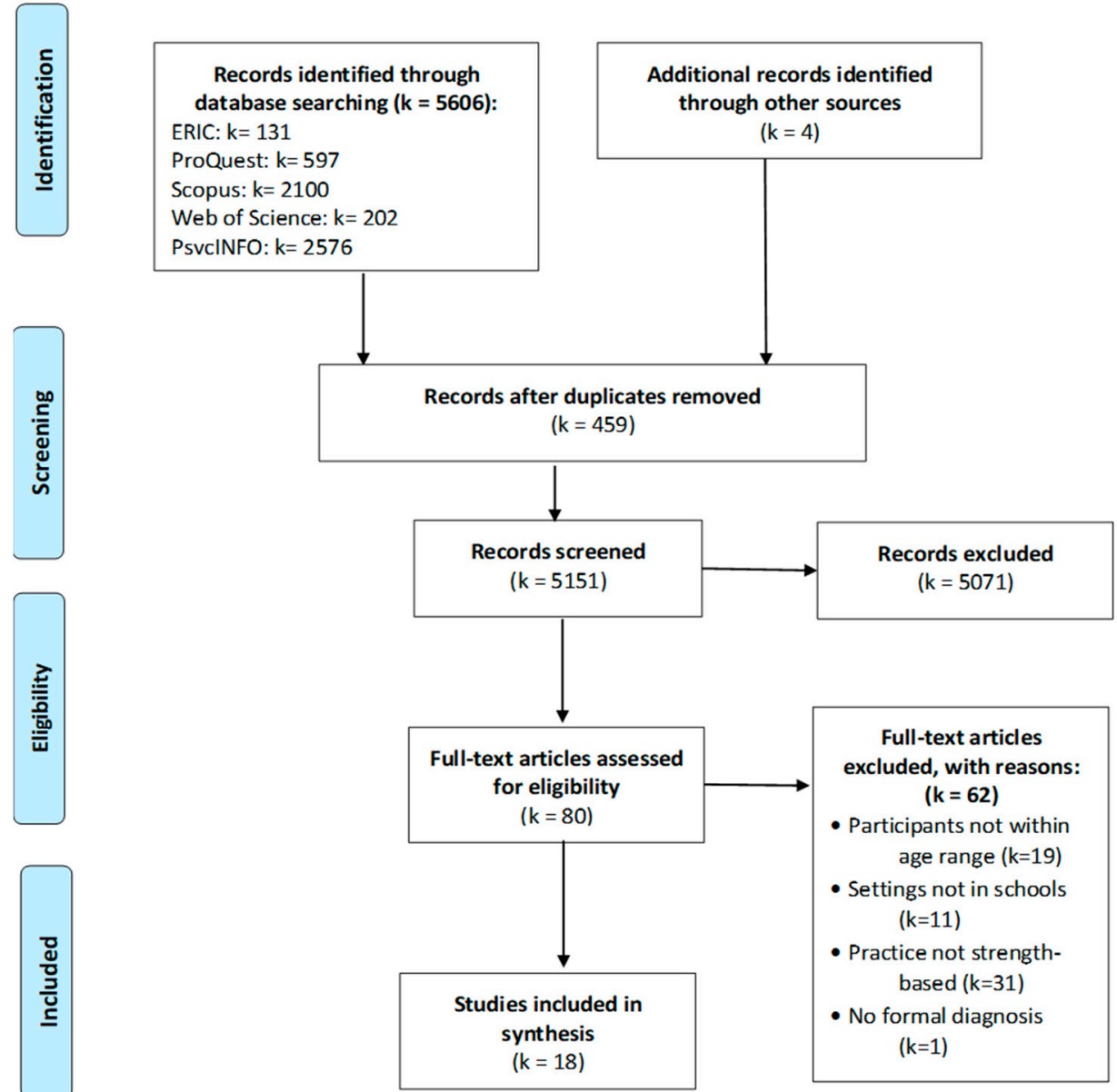

**Figure 2.** Flowchart of study selection process.

*3.2. Study Design*

The 18 selected articles were published between 2009 and 2022, with the majority being from the United States (k = 15), two from Australia (k = 2), and one from the United Kingdom (k = 1). Thirteen out of the eighteen articles were research articles, with six studies investigating the different stakeholders' perspectives and experiences of strengths-based approaches for autistic students, utilising qualitative (k = 4), quantitative (k = 1) or mixed-methods (k = 1) design. Seven studies evaluated strengths-based interventions and strategies for autistic students in school settings, applying methodologies including multiple-baseline design (k = 4), alternating treatment design (k = 1), and one each adopting a qualitative design (k = 1) and longitudinal mixed methods design (k = 1). Another five articles were practitioner-based and did not report original research but instead focused on translating research into actionable practices and were included because the actionable guidance and examples were valuable for educators implementing evidence-based practices [7,36–39] (Table 2).

**Table 2.** Descriptive features of included studies (*k* = 18).

| Author (Year), Country | Design | Participants/Referenced Population | Context | Objective/ Research Question | Outcome Measures | Outcomes | Targeted Areas | Methodological Quality |
|---|---|---|---|---|---|---|---|---|
| Bargerhuff (2013) [40] USA | Qualitative Single site case study | School staff of students with disabilities, including autism; *n* = 9 (2 males; 7 females) including 6 teachers, 1 principal, 2 special educators, and 1 administrative assistant | A STEM public high school | 1. What are the primary supports to the learning of SWD in this school? 2. What are the primary challenges to the learning of SWD in this school? 3. How does the working relationship among professionals in this school support/hinder the learning of SWD? | Interviews Observation Document examination | Staff in the STEM school take ownership and maintain high expectation to meet all students' individual learning needs through inquiry project-based learning, technology and collaboration | Academic, transitional | 16/20 80% strong |
| Bellini and McConnell (2010) [7] USA | Practitioner-based paper | Autistic elementary school student (*n* = 1) | School and classroom | To summarise research investigating VSM and to identify common obstacles and challenges for teachers to implement VSM for autistic in schools; To provide practical examples of VSM interventions that required minimal time and skills for teachers. | N/A | Supported skills are transformed into independent skills through the use of VSM | Social, behavioural | N/A |
| Bianco et al. (2009) [36] USA | Practitioner-based paper | Gifted autistic student; *n* = 1 (male), Age: 9 (Grade four) | School and classroom | To offer suggestions and resources for developing strengths-based programming for gifted students on the autism spectrum. | N/A | Gifted students on the autism spectrum are supported through interventions that foster their interests and strengths while providing strategies to support their weaknesses | Academic, social and emotional | N/A |



**Table 2.** *Cont.*

| Author (Year), Country | Design | Participants/Referenced Population | Context | Objective/ Research Question | Outcome Measures | Outcomes | Targeted Areas | Methodological Quality |
|---|---|---|---|---|---|---|---|---|
| Bottema-Beutel et al. (2016) [10] USA | Mixed-methods | Autistic youth; *n* = 33 (33% female, 67% male), Age: 14–25, M = 17.8 | School | To investigate the viewed favourability by autistic youth of the seven school-based, social-focused, and peer-mediated intervention components | Rating of the favourability of components of social-focused, peer-mediated interventions Interviews (in-person, video chat, phone, mail, email, and instant messaging) | The preferences of autistic youth in school-based social interventions are investigated | Social and emotional | 94% Strong (19/20, 95% quantitative; 18/20, 90% qualitative) |
| Bross and Travers (2017) [37] USA | Practitioner-based paper | Autistic student in senior high school; *n* = 1 (male) | High school | To propose a four-step process to provide school-based opportunities aligned with the SIA of an autistic student to improve their employment skills | N/A | Increased interests in the class; better communication skills; developing employment skills | Transitional; planning and employment skill training | N/A |
| Chalfant et al. (2017) [38] USA | Practitioner-based paper | Autistic students (sample size, gender and age not reported) | Classroom | To address evidence-based practices that can be embedded in science classes | N/A | Enhanced access to Science content | Academic, social and emotional, behavioural | N/A |
| Chen et al. (2022) [41] USA | Qualitative research | 6th–8th grade autistic students, *n* = 17 (14 males, 3 females); non-autistic students, *n* = 9; parents of the autistic students, *n* = 13; teachers implemented the program, *n* = 9; | Three public middle schools | To explore the experiences and perceived outcomes of students, teachers and parents participating in an inclusive, school-based informal engineer education program | Interviews, focus groups, program implementation logs/field observation notes | Positive student experience, skills and interest development in STEAM and related careers, enhanced social relationships and self-determination | Academic, social and emotional, transitional | 18/20 90% Strong |

| Author (Year), Country | Design | Participants/Referenced Population | Context | Objective/ Research Question | Outcome Measures | Outcomes | Targeted Areas | Methodological Quality |
|---|---|---|---|---|---|---|---|---|
| Davis et al. (2010) [42] USA | Multiple-baseline design | Autistic high school students; *n* = 3 (100% male), Age: 16, 17, 17 (Grade 11 and 12); peers without disabilities participated as conversational partners; *n* = 20 | A special education resource room and a conference room in general education settings | To evaluate the use of the Power Card strategy on conversation skills for autistic high school students. | Observation of the time the autistic students maintained allowing the conversational partners to speak about their interests. Social survey regarding social validity | Increased percentage of time engaged in other-focused conversations | Social and emotional | 13/20 65% Adequate |
| Holcombe and Plunkett (2016) [43] USA | Qualitative research case study | Educators with various roles in the public education sector who have had a close working relationship with at least one autistic student within the past 12 months; *n* = 56 (6 specialists, 15 classroom teachers, 11 coordinators or team leaders, 9 education support officers, 8 assistant principals and 7 principals) | 28 in the primary sector, 19 in the secondary sector, 4 in the specialist sector and 5 across primary, secondary and specialist settings) | How can support autistic students be more effectively understood, implemented and experienced in mainstream schools? | Online questionnaire; Semi-structured interviews | Student outcomes and achievement; well-being, engagement, planning and programming, positive school community | Academic, social and emotional, behavioural | 11/20 55% Adequate |

**Table 2.** *Cont.*

| Author (Year), Country | Design | Participants/Referenced Population | Context | Objective/ Research Question | Outcome Measures | Outcomes | Targeted Areas | Methodological Quality |
|---|---|---|---|---|---|---|---|---|
| Koegel et al. (2012) [44] USA | Quantitative; repeated measures; multiple baseline experimental design | Autistic students; *n* = 3 (males), Age: 11–14 | Local junior and senior high schools | To systematically assess the effectiveness of structured lunchtime clubs that were organised and themed based on the perseverative interests of high school autistic students regarding their social interactions with neurotypical peers. | Engagement or initiations | Large increases in both social engagement and initiations | Social and emotional | 13/16 81% Strong |
| Koegel et al. (2013) [45] USA | Repeated multiple-baseline across participants design | Autistic students; *n* = 7 (six males, one female), Age: 14–16 | Lunchtime in inclusive high school settings | To understand how by incorporating their preferred interests, to enhance the engagement of autistic students with neurotypical peers, including initiations made to typical peers, during social activities in an inclusive high school | The percent intervals of autistic adolescents' engagement; rate of initiations the adolescent with made to neurotypical peers; social validation measures of self-reports from autistic and non-autistic adolescents | Increases in both level of engagement and rate of initiations made to neurotypical peers | Social and emotional | 18/20 90% Strong |
| Koegel et al. (2018) [46] USA | Quantitative alternating treatment experimental design | Autistic students; *n* = 2 (males), Age: 8 and 91/2 | Lunch time or recess periods in two public elementary schools | To understand how activity history may influence socialization and engagement during activities that incorporated restricted repetitive behaviours of autistic students. | Activity engagement Social engagement Initiations to peers | Socialization increased and remained above baseline levels when RRBs were introduced during activities with a positive history | Social and emotional | 16/22 73% Good |

Table 2. *Cont.*

| Author (Year), Country | Design | Participants/Referenced Population | Context | Objective/ Research Question | Outcome Measures | Outcomes | Targeted Areas | Methodological Quality |
|---|---|---|---|---|---|---|---|---|
| Lanou et al. (2012) [39] USA | Practitioner-based paper | Autistic students; *n* = 4 (males), Grade five | Inclusive classrooms | To present strategies developed for autistic students that capitalise on the students' authentic interests and strengths to meet their school-based challenges | N/A | Increased writing stamina and productivity; better communication of feelings; decreased intensity of frustration and improved recovery time; better self-monitoring skills and use of calming strategy Decrease in invading peers' space; better understanding of personal space. | Academic (writing), social and emotional, behavioural | N/A |
| Maras et al. (2019) [47] UK | Quantitative 2 × 2 between-participant design | Autistic students, *n* = 40 (30 males, 10 females), Age: 11–16, Mean = 13.33 years (SD = 1.25); Comparison participants neurotypical secondary school students, *n* = 95 (58 males, 37 females), Age: 11–15, Mean age = 13.4 (SD = 1.15) | Specialist provision classroom within mainstream schools | To test a computer-based metacognitive support (the 'Maths Challenge') for mathematics autistic learners in classrooms. | Pre-test intention measure; post-test metacognitive monitoring confidence judgement; post-test intention measure | Undiminished ability to detect errors with reduced cohesion between pre- and post-test intentions | Academic (meta-cognition skills–self-regulation) | 14/20 70% Good |
| McKenney et al. (2016) [48] USA | Exploratory, observation-based study | Autistic middle and high school students, *n* = 16, age: 12–18; | Secondary general education settings | To strengthen understanding of the development of social communication skills that facilitate academic success, particularly within general education settings. | Exploratory, observation-based study | Autistic students were more likely to engage in appropriate, facilitative behaviours with the classroom setting. | Academic, social and emotional | 18/20 90% Strong |

**Table 2.** *Cont.*

| Author (Year), Country | Design | Participants/Referenced Population | Context | Objective/ Research Question | Outcome Measures | Outcomes | Targeted Areas | Methodological Quality |
|---|---|---|---|---|---|---|---|---|
| Shochet et al. (2022) [49] AUS | Longitudinal mixed methods study | Autistic adolescents in Years 7 and 8, *n* = 30 (24 males, 6 females), Age: 11–14 (Mean = 11.84 SD = 0.86); their parents, *n* =31; teachers, *n* = 16 | Six secondary schools | To evaluate the feasibility and outcomes of multiple ecological level school-based resilience and mental health intervention program | Pre- and post-tests measures; semi-structured interviews | Autistic adolescents showed an increase in resilience, affect regulation, a sense of belonging and coping self-efficacy | Social and emotional | 93% Strong (20/22 91% Quantitative; 19/20 95% Qualitative) |
| Stokes et al. (2017) [50] AUS | Qualitative ground theory approach | Principals, *n* = 29 (13 males, 16 females) and teachers, *n* = 29 (6 males, 23 females), of autistic students | 18 primary schools and 11 secondary schools | To collect perspectives of principals and teachers on successful teaching strategies with autistic students in classroom environments and educational settings. | Online survey and online reflective journal | Both teachers and principals found numerous strategies such as structure and incorporating needs to be successful. Inappropriate communication, disorganisation, and a lack of understanding were unhelpful | Academic, social and emotional, behavioural | 16/20 80% Strong |
| Winter-Messiers et al. (2007) [51] USA | Qualitative interviews and surveys | Autistic students, *n* = 23, Age: 7–21 years; parents; *n* = 18 | School | To understand how the SIAs in autistic youth origin and develop, and their experiences related to their SIAs at school. | All children displayed enhanced functioning in one or more of their deficit areas when they were engaged in describing their SIAs. | Increased communication, social, emotional, sensory, fine-motor, executive function and academic skills when engaging in special interest areas | Social and emotional, academic, behavioural, transitional | 7/20 35% Limited |

STEM: Science, Technology, Engineering and Mathematics; SWD: students with disabilities; VSM: Video-Self Modelling; RI: restricted interests; SIA: special interest area; RRB: restricted repetitive behaviour.

### 3.3. Methodological Quality of Included Studies

The research quality of the 13 primary studies ranged from limited (k = 1), to adequate (k = 2), to good (k = 2) to strong (k = 8) (Table 2). Common strengths in all studies include a comprehensive description of the research question and design and reporting of results in sufficient detail. Common limitations in the quantitative studies include inappropriate sample size and lack of detailed report on analytic methods. Only two out of the seven intervention studies reported on statistical significance [47,49]. Common limitations in the qualitative studies include limited description of data analysis, limited verification procedure to establish credibility and no reflexivity of the account. The practitioner-based articles (k = 5) were not assessed for research quality because the assessment criteria for evaluating primary research papers [34] do not apply to this type of articles, due to their research-to-practice focus rather than reporting original research [52].

### 3.4. Participant Groups Identified within the Studies

Participants in the seven intervention studies included autistic students (*n* = 102, male *n* = 82, female *n* = 20) ranging from 8 years old to 17 years old (mean = 13.34), and one study did not report age ranges, only grade levels (grades six to eight) [41]. Three studies included only male students [42,44,46], and four studies contained both male and female students [41,45,47,49]. Among the six studies investigating the different perspectives and experiences, three studies examined the views of autistic students and youth (*n* = 72) [10,48,51], with one study including the views of parents (*n* = 18) on the types of strengths autistic students have [51], and three studies investigated the experiences and perspectives of school staff (*n* = 123), including teachers, principals, education support officers and administrative assistants [40,43,50]. The practitioner-based papers did not report participants but described autistic students ranging from primary school [7,36,39] to senior high school [37,38] as examples to explain the implementation of the research-based strategies.

### 3.5. Outcomes of Strengths-Based Approaches in School

Strengths-based approaches have been suggested to support the social and emotional, academic, behavioural and transitional outcomes of autistic students (Table 2). All articles except one [47] targeted emotional and social outcomes (k = 17). Some authors [36,39,44,45,51] hypothesised that strengths-based approaches may assist in enhancing skills in expressing, monitoring and managing one's emotions. Koegel et al. [44] and Koegel et al. [45] reported improved frequency in initiating and maintaining conversations and increased feelings of happiness and enjoyment in autistic students following implementing lunchtime activities and clubs incorporating their interests in high school settings. Shochet et al. [49] implemented a strengths-focused multilevel intervention at school that was found to improve students' social, emotional resilience and mental health. Ten studies utilised students' strengths and interests to improve academic outcomes, from improved academic skills in one specific area, such as writing skills [39] to academic outcomes in Science, Technology, Engineering and Mathematics (STEM) areas [38,40,41,47] and overall academic performance [36,43,48,50,51]. Seven articles reported teacher observed increased on-task behaviours, improving focus and completion, and developing self-management and independence [7,38–40,43,50,51]. Four articles considered benefiting transitional outcomes, including aiming for some form of post-secondary education [40], and linking their special interest areas (SIAs) to early employment experiences at school and transition planning for a meaningful job, but did not report the measured outcomes [37,41,51].

### 3.6. Key Elements for Strengths-Based Approaches in School Environments

Through the narrative synthesis approach [33] and using Bronfenbrenner's Bioecological Model of Development as a framework, key elements relating to the implementation of strengths-based approaches are identified and organised into one person related element, strengths and interests, and twelve elements within the different layers of school envi-

ronment including six elements in the microsystem (specialised instructions, curriculum integration, curriculum differentiation, common interests with peers, reciprocal roles and adult involvement), three elements in the mesosystems (matching resources and activities, real-life learning experiences and benefiting all students), and three elements in the exosystem (cost-effective and time-saving, collaboration with colleagues and parents and teachers' attitude and knowledge) (Figure 3). Articles contributing to the person related element are listed in Table 3, and articles contributing to the environmental elements are listed in Table 4.

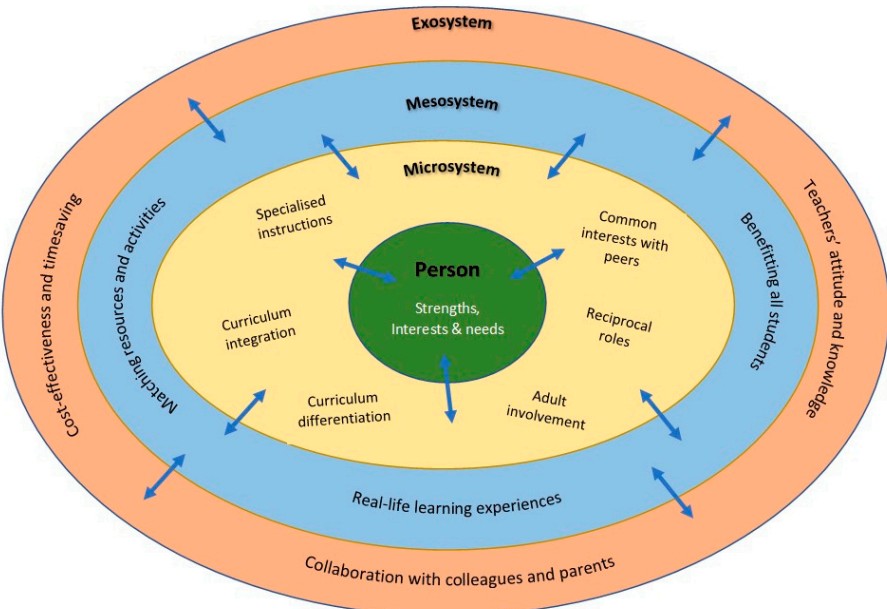

**Figure 3.** Person and environmental elements contributing to strengths-based approaches for autistic students in schools.

### 3.6.1. Person-Related Element

- Strengths and interests

All articles except two [10,49] identified at least one specific strength or interest of autistic students (Table 3). The strengths relating to students' learning and school experiences include cognitive abilities, knowledge in specific subjects linked to their SIAs, visual and sensory processing abilities, music, art, sports and motor skills and personality and attitude. The common interests include computer and video games, vehicle and transportation, sports, cooking and comics and cartoons. Nine articles reported methods of identifying students' strengths and interests to achieve individualised planning of strengths-based approaches. Using a combination of preference assessments, observations and interviewing students and parents was considered more effective than only relying on one assessment method [36,37,39,43,51]. Actively involving the students themselves and being flexible was emphasised by Bargerhuff [40] and Stokes et al. [50] as important to individualised planning. Bross and Travers [37] and McKenney et al. [48] both found that to ensure programmatic use of students' strengths and interests, consistently monitoring students' progress and adjusting the strategies accordingly through formative assessments is needed.

**Table 3.** Strengths and interests identified in the articles.

| Author and Year | Strengths | | | | | | | Interests | | | | | |
|---|---|---|---|---|---|---|---|---|---|---|---|---|---|
| | Cognitive and Metacognitive Abilities | Visual Processing and Sensory Processing | Music, Drama and Art | Social Verbal, Vocabulary and Communication | Attention to Detail and Organisation and Executive Function | Sports and Motor Skills | Sense of Humour, Motivation, Commitment and Willing to Learn | Computer and Video Games | Knowledgeable in Specific Subjects | Vehicles and Transportation | Sports | Cooking | Comic and Cartoon and Movies |
| Bargerhuff (2013) [40] | ▪ | | | | | | | | | | | | |
| Bellini and McConnell (2010) [7] | | ▪ | | | | | | | | | | | ▪ |
| Bianco et al. (2009) [36] | | ▪ | | ▪ | | | | | ▪ | | | | |
| Bottema-Beutel et al. (2016) [10] | | | | | | | | | | | | | |
| Bross and Travers (2017) [37] | | | | | | | | | | ▪ | | | |
| Chalfant et al. (2017) [38] | | ▪ | | | ▪ | | | | | | | | |
| Chen et al. (2022) [41] | | | | | | | | ▪ | ▪ | | | | |
| Davis et al. (2010) [42] | | | | ▪ | | | | | | | ▪ | | ▪ |
| Holcombe and Plunkett (2016) [43] | ▪ | ▪ | ▪ | ▪ | ▪ | ▪ | ▪ | | | | | | |
| Koegel et al. (2012) [44] | | | | | | | | | | | ▪ | | ▪ |
| Koegel et al. (2013) [45] | | | | | | | ▪ | ▪ | | | ▪ | ▪ | ▪ |
| Koegel et al. (2018) [46] | | | | | | | | ▪ | | ▪ | | | |
| Lanou et al. (2012) [39] | ▪ | ▪ | | ▪ | | | | | | | | | |
| Maras et al. (2019) [47] | ▪ | | | | | | | | | | | | |
| McKenney et al. (2016) [48] | ▪ | ▪ | | ▪ | ▪ | ▪ | | | | | | | |

**Table 3.** *Cont.*

| Author and Year | Strengths | | | | | | | | | | Interests | | |
| --- | --- | --- | --- | --- | --- | --- | --- | --- | --- | --- | --- | --- | --- |
| | Cognitive and Metacognitive Abilities | Visual Processing and Sensory Processing | Music, Drama and Art | Social Verbal, Vocabulary and Communication | Attention to Detail and Organisation and Executive Function | Sports and Motor Skills | Sense of Humour, Motivation, Commitment and Willing to Learn | Computer and Video Games | Knowledgeable in Specific Subjects | Vehicles and Transportation | Sports | Cooking | Comic and Cartoon and Movies |
| Shochet et al. (2022) [49] | | | | | | | | | | | | | |
| Stokes et al. (2017) [50] | | ▪ | | | | | | ▪ | | | | | |
| Winter-Messiers et al. (2007) [51] | ▪ | | | | | | | | | | | | |

**Table 4.** Article contribution to environmental elements.

| Author (Year) | Elements | | | | | | | | | | | |
| --- | --- | --- | --- | --- | --- | --- | --- | --- | --- | --- | --- | --- |
| | Microsystem | | | | | | Mesosystem | | | | Exosystem | |
| | Specialised Instructions | Curriculum Integration | Curriculum Differentiation | Common Interests with Peers | Reciprocal Roles | Adult Involvement | Matching Resources and Activities across the School | Real-Life Learning Experiences | Benefiting All Students | Cost-Effective and Time Saving | Collaboration with Colleagues and Parents | Teachers' Attitude and Knowledge |
| Bargerhuff (2013) [40] | ▪ | ▪ | ▪ | ▪ | ▪ | ▪ | | ▪ | ▪ | ▪ | ▪ | ▪ |
| Bellini and McConnell (2010) [7] | ▪ | | | | | | | | | ▪ | | ▪ |
| Bianco et al. (2009) [36] | ▪ | ▪ | ▪ | | | | | ▪ | | ▪ | ▪ | ▪ |
| Bottema-Beutel et al. (2016) [10] | ▪ | | | ▪ | ▪ | ▪ | | ▪ | | | | |
| Bross and Travers (2017) [37] | ▪ | | | | | | ▪ | ▪ | | | ▪ | ▪ |

**Table 4.** *Cont.*

| Author (Year) | Elements | | | | | | | | | | | |
|---|---|---|---|---|---|---|---|---|---|---|---|---|
| | Microsystem | | | | | | Mesosystem | | | | Exosystem | |
| | Specialised Instructions | Curriculum Integration | Curriculum Differentiation | Common Interests with Peers | Reciprocal Roles | Adult Involvement | Matching Resources and Activities across the School | Real-Life Learning Experiences | Benefiting All Students | Cost-Effective and Time Saving | Collaboration with Colleagues and Parents | Teachers' Attitude and Knowledge |
| Chalfant et al. (2017) [38] | ■ | | ■ | | | ■ | | | ■ | | ■ | ■ |
| Chen et al. (2022) [41] | ■ | ■ | | ■ | ■ | | ■ | ■ | ■ | | | ■ |
| Davis et al. (2010) [42] | ■ | | | | | | | | | ■ | | ■ |
| Holcombe and Plunkett (2016) [43] | ■ | | | | | | | | ■ | | | ■ |
| Koegel et al. (2012) [44] | | | ■ | ■ | | ■ | ■ | | | | | |
| Koegel et al. (2013) [45] | | | | ■ | | | ■ | | ■ | | | |
| Koegel et al. (2018) [46] | | | | | | | ■ | | | | | |
| Lanou et al. (2012) [39] | ■ | | ■ | | | | | | | | | ■ |
| Maras et al. (2019) [47] | | | ■ | | | | | | | ■ | | |
| McKenney et al. (2016) [48] | | | | | | | | | | | | |
| Shochet et al. (2022) [49] | ■ | | | | | | | ■ | | | ■ | ■ |
| Stokes et al. (2017) [50] | ■ | | ■ | | | | | | ■ | ■ | | ■ |
| Winter-Messiers et al. (2007) [51] | | | ■ | | | | | | | | ■ | ■ |

3.6.2. Elements within the Microsystem

Six elements, including specialised instructions, curriculum integration, curriculum differentiation, common interests with peers, reciprocal roles and adult involvement, were identified in the microsystem, which is a student's immediate environment with teachers and peers.

- Specialised instructions

Specialised instructions based on a student's strengths and interests were utilised to help autistic students access the information they need [7,36–43,49]. Given that visual processing was identified as a strength of many autistic students, visual supports were commonly used. Visual supports included the use of diagrams, graphs, concept maps, timelines, outlines and photographs, with typical examples including Video-Self Modelling (VSM) and social storying. Several studies found that the use of visual supports assisted in making instructions easier to understand for autistic students and provide visual reminders for students, improving understanding of both the curriculum content and social contexts and social rules [7,36–38,42]. Visual cues were also found to act as a tool to provide activity schedules and visual reminders building self-regulation and metacognitive skills that are important for both autistic students' current behavioural, social and emotional and academic needs, and their transition to adulthood [7,39–41,43,49].

Another type of specialised instruction for autistic students is priming. Priming was suggested to be provided prior to classes for preparing autistic students individually for lessons and reducing the need for additional prompting or modification [38,39,50]. Chalfant et al. [38] suggested that priming can also support autistic students in group activities to provide direct instructions regarding how to share interests and engage in conversations with peers. Providing explicit instructions to support the participation of autistic students is important in social-focused, peer-mediated interventions, as Bottema-Beutel et al. [10] reported that the majority of autistic youth participating in their study preferred group activities with peers only when it was easier or more enjoyable.

- Curriculum integration

Three articles reported developing themes across different curriculum areas to utilise and leverage autistic students' interests and strengths [36,40,41]. Bargerhuff [40] found that the programs with different themes integrating different curriculum areas implemented in a STEM school were effective for motivating all students, including autistic students, because the content was relevant to students' interests and lives, and represented teachers' high expectations of students' outcomes. In one study, an inclusive, school-based engineering design program, the IDEAS Maker Program, required students to undertake projects drawing on knowledge and skills from several different areas, designed to reflect the real-life experiences [41]. Bianco et al. [36] suggested that teachers should brainstorm with the students themselves to expand areas of study around the students' interests. These areas can then be developed into interdisciplinary thematic units including Art, Mathematics, Literature, History and Cultural studies, which in turn create opportunities to broaden the student's interests. Exploring learning opportunities through curriculum integration can also generate ideas for other resources within the school to be included and applied [36].

- Curriculum differentiation

Eight studies examined how curriculum could be differentiated to address some of the challenges based on the strengths and interests of these students [36,38–40,44,47,50,51]. Bianco et al. [36] used the term "curriculum dual differentiation" to refer to the programming that considers both the abilities and limitations of the students. Winter-Messiers et al. [51] provided examples of integrating SIAs into the core academic areas for autistic students to motivate them in learning by allowing them to showcase their real levels of academic abilities. For instance, using their interest in engaging the internet in non-preferred topics or challenging learning areas and completing tasks to earn free time for their interests. Curriculum differentiation also includes modifications on the expected learning outcomes

and assessments. Providing flexibility, including extra time and alternatives for taking tests, having access to computer technology for reading, writing and organization and use of social cues to guide behaviour during class were also being considered to provide autistic students with alternative pathways to access curriculum and demonstrate their learning [40,50].

- Common interests with peers

　　Five studies found that developing peer relationships based on common interests provided peers with a focal point topic for communication and a structure to follow. It also promotes natural interactions through activities in a safe environment [10,40,41,44,45]. This was found to be particularly important for autistic adolescents for generalising the skills outside of the school environment and developing lasting friendships [10,44].

- Reciprocal roles

　　Three articles reported on the importance for autistic students and their peers to play reciprocal roles in peer-mediated learning. Bargerhuff [40] reported that students with disabilities frequently volunteered to lead in activities such as Socratic circles and dialogue sessions. Chen et al. [41] found that peer teaching in the IDEAS Maker Program promoted relationship and community building and developed a safe environment supporting self-determination in all students. Bottema-Beutel et al. [10] also reported the autistic high school students' view that they would like to be understood and treated equally by their neurotypical peers and therefore expressed mixed feelings about revealing their autism diagnosis with other students. This was deemed by autistic youth to be an important factor in social skill interventions.

- Adult involvement

　　Studies discussed two sides of adult involvement in peer-mediated learning for autistic adolescents. First, both autistic and neurotypical students need to be provided essential support for peer-mediated learning to work. For instance, studies reported that all students need to be taught explicit strategies for effective communication, sharing and clarifying content and organising information. Additionally, roles and expectations need to be clearly defined [38,40]. Meanwhile, Koegel et al. [44] and Bottema-Beutel et al. [10] both emphasised that adult involvement, whether it is from teachers or parents, should be through an indirect approach, for instance, providing logistic assistance and resources, instead of setting goals from an adult's perspective. Arranging and promoting interactions with peers is another way to support through an indirect approach. The adult involvement should fade slowly during the process to allow youth interactions to progress naturally.

3.6.3. Elements within the Mesosystem

　　Three elements were identified in the mesosystem representing connections between the different settings involving the autistic student. The three elements are matching resources and activities, real-life learning experience and benefiting all students.

- Matching resources and activities across the school

　　Because of the wide range and variety of students' needs, after identifying the SIAs and strengths, matching them to resources and activities was identified as one way to individualise planning and to implement the interventions to meet the individual needs of autistic students. Bross and Travers [37] suggested the use of schedules to identify the possible available locations, people and resources across the school to integrate students' SIAs into the existing daily activities and academic learning. For extra curriculum activities, studies reported modifying the themes, topics and format of the current student club activities to incorporate the students' interests and strengths and to best fit within the school environment to improve their social engagement and initiations with peers [41,44–46].

- Real-life learning experiences

Providing real-life learning experiences was another element within strengths-based approaches. Real-life learning means solving real-life problems and evaluating students' abilities and learning outcomes in authentic, real-world contexts [36,40,41,49]. Studies found that autistic students preferred natural interactions with peers through activity-based learning incorporating themes of their perseverative interests, which also promotes generalisation of the developed social skills and friendship outside of school-based interventions [10,41,44]. Autistic adolescents also reported becoming more resourceful and noticed positive changes in themselves when they could apply what they learned in a school-based mental health intervention program in their daily life [49]. Real-life learning also creates early work-related experiences at school by capitalising on student SIAs, which contributes to their long-term goals [37,41].

- Benefitting all students

Ensuring that programs and strategies benefit all students while supporting autistic students is important in inclusive mainstream schools. Holcombe and Plunkett [43] proposed the Bridges and Barriers Model of Support (BBMS) for autistic students by focusing on universal barriers for all students rather than only addressing autism specific characteristics. Stokes, Thomson [50] also reported that many of the strategies that are reported by teachers and principals to be effective for autistic students were also considered good practice for all students. When supporting autistic students' participation in group work, providing all students with explicit instructions on strategies of collaboration and organisation also benefit all students [38,40]. While in school-based clubs and programs incorporating autistic students' SIAs, their neurotypical peers also reported their enjoyment of the experience [41,45].

3.6.4. Elements within the Exosystem

Three elements, cost-effectiveness and time-saving, collaboration with colleagues and parents and teacher professional development are identified within the exosystem of school environment that enable strengths-based approaches for autistic students.

- Cost-effectiveness and time-saving

Cost-effectiveness and time-saving are crucial features of a strategy or intervention that encourages teachers' implementation, as a lack of time and support was identified by teachers as a barrier to providing support for autistic students [7,50]. For instance, Video-Self Modelling (VSM), allowing students to imitate targeted behaviours by observing themselves successfully performing the behaviour with or without additional support pre-recorded in a short video, was considered an efficient and effective strengths-based instructional strategy [7]. Davis, Boon [42] found that using Power Cards, small, laminated cards containing visual and text script of conversation strategies used by a character of the students' choice, improved autistic high school students' engagement in conversations. Power Cards were considered inexpensive and easy to make and can be carried around and used by the students themselves after the initial training. Technology such as dictation software programs to compensate for deficits in writing, text-to-speech devices, using internet to access distance mentoring and computer-based metacognitive support programs, may also provide tools for inexpensive and efficient programs and strategies based on students' interests and strengths for teachers to utilise [36,40,47].

- Collaboration with colleagues and parents

Six studies considered colleagues and parents as resources for successful planning and implementing strengths-based practice [36–38,40,49,51]. Collaboration between general educators and special education teachers was deemed important because special education teachers may have specialised knowledge regarding instruction in social skills and differentiating instruction that will be of benefit [38,40,51]. Bianco et al. [36] also suggested to interview parents for their child's strengths, interests and needs because autistic students often pursue their interests and demonstrate their leadership outside of school, while Bross

and Travers [37] suggested finding experts and resources in communities for developing employment skills programs on the special interest areas of autistic students. To promote mental health and wellbeing in autistic adolescents, one study included components for parents and teachers as well as for students in their school-based intervention program [49].

- Teachers' attitude and knowledge

Nine papers reported the positive or negative impact of teachers' attitude, perceptions and knowledge related to strengths-based practice. Bross and Travers [37] and Lanou et al. [39] reported that for the successful implementation of the strengths-based practice, teachers need to recognise powerful effects of students' interests and strengths, rather than viewing them as a deficit to be changed. In the study of Bargerhuff [40], teachers' appreciation of students' strengths, acceptance of their differences and ownership of learning is considered a crucial factor to the success of strengths-based approaches in a STEM high school. The findings of Holcombe and Plunkett [43] also show that it is important for educators to move beyond implementing strategies designed by others and to develop expertise in designing what is effective for their students and their teaching in their own circumstances.

Two studies [7,50] identified that a teacher's lack of knowledge of how to identify the needs of students and their strengths and interests as a barrier to the strengths-based practice as they do not know how they can better support their students even though they might want to. In a study investigating the views and experiences of teachers and principals [50], a lack of understanding of autism by teachers was also considered a barrier to effective teaching. Teacher perceived difficulty of implementation, lack of time and ability to access new technology also reduced teachers' willingness to utilise technology-based strategies, such as the VSM [7].

Four articles explored teacher professional development and training. Providing teachers with training on the needs of autistic students and teaching strategies was an important component both in the school-based IDEAS Maker Program [41] and mental health intervention program [49]. Chalfant et al. [38] encouraged teachers to seek appropriate training for the effective implementation of research-based practices and strategies considering the wide range of needs autistic students have, and there is no single best method or approach for all of them. Whilst Stokes et al. [50] reported that information and advice from specialists, as well as through internal or external professional development opportunities are the main sources to inform teachers' practices compared to information from parents, students and internet.

## 4. Discussion

This review examined strengths-based approaches for autistic students in schools, using the Bioecological Model of Development to identify the key individual and environmental factors important for the implementation of strengths-based approaches in school. Although strengths-based programs have been applied in some community-based interventions with positive outcomes for autistic students [22,53,54], it is evident from this review that there is still a paucity of research on strengths-based approaches in schools, especially for older students in high school settings. Since schools are one of the microsystems where students spend the majority of their time, and the purpose of interventions is to improve autistic students' abilities to participate in real-world experiences, schools should be an ideal place for making changes for autistic adolescents [55,56]. The positive outcomes that were observed from strengths-based approaches by papers in the current review support that schools should continue to move away from traditional practices and operating from a deficit-based model dedicated to "fixing" the students, to practices that recognise and leverage their strengths and interests to empower the students [7,37,57].

This review found that strengths-based approaches would benefit from addressing the elements within an individual student and across the different layers of the school environment. Strengths-based approaches do not ignore the challenges and difficulties that autistic students face, rather, they empower students themselves to build their com-

petency to address the challenges and difficulties by focusing on what they can do and what they are good at and providing the support in the environment that is sympathetic to their style [58]. At the individual student level, studies in this review identified many strengths and interests that autistic students have as a person-related element contributing to strengths-based approaches. The strengths and interests identified by the studies included in this review have added further evidence to the various positive traits, interests and strength-related profiles reported by autistic children and youth themselves [59,60] and by their parents [61]. As these strengths and interests are usually not demonstrated in the typical academic achievement and behavioural data, utilising strengths assessments to provide measurements on the strengths of students as well as educators is an important starting point of strengths-based approaches [16]. These strengths, including abilities, knowledge, skills and experiences as well as interests, which Bronfenbrenner [23] described as resource and force characteristics, can be leveraged in the school environment to address the needs of these student to optimise their social, emotional, behavioural, academic and transitional outcomes.

The elements identified in the microsystem demonstrate the importance in differentiating the content, structure and assessment of learning and provide instructions that specifically work for autistic students so they can flourish at school. Educators making an effort to think about and act upon each student's strengths to individualize their learning experiences is also considered by Lopez and Louis [16] as one of the principles of strengths-based education. However, it is evident that there are more studies focusing on specialised instructions and curriculum differentiation. Only three articles [36,40,41] considered curriculum integration, which relies more heavily on resources and activities across the different sectors of schools, real-life learning experiences and teacher collaboration, elements in the mesosystem and exosystem. This indicates that to provide rigorous learning opportunities for autistic students, more research and practices need to be explored in this area. For example, future research and practice may benefit from applying approaches related to Universal Design for Learning [62], by considering the WHY of learning, the WHAT of learning and the HOW of learning of each student to design learning based on their strengths and interests to achieve inclusivity and holistic development.

Quality relationships were found to have a significant contribution in the successful implementation of strengths-based approaches across the different levels of school environment. Within the microsystem, both peer relationships and adult involvement make a difference in the participation of autistic students. The common interests provided opportunities and structures for autistic students to communicate with their peers, to participate in school activities and to develop their competence and generalisation of skills in authentic, real-time contexts and settings. However, only two studies looked at the reciprocal roles that autistic students play based on their strengths. Future strengths-based approaches can explore more opportunities to intentionally develop autistic adolescents' strengths by considering their contribution to the group and peers to increase their self-esteem, social satisfaction and sense of belonging [9,16].

Building relationships among the different stakeholders involved in the strengths-based practice is also relevant to all of the elements within both the mesosystem and exosystem. Benefitting all students is particularly important for making autistic students feel included and not isolated, which is crucial for adolescents. It also makes it easier for teachers to implement the practice in an inclusive school environment, enhancing social and educational outcomes for all students [63]. The element of collaboration among the adults, including teachers, other professionals and parents also links to the element of matching resources and activities across the school, both are essential for strengths-based approaches [64], as applying a team approach is essential in planning and implementing actions and activities in schools [65]. To further future research and practice into connecting all these elements across the different layers of the ecological system, the Index for Inclusion [66] can be utilised as a tool for promoting inclusion in education by considering all three dimensions, cultures, policies and practices, of a school. Only one study [49]

included in this review aligned the intervention with the Index for Inclusion, but with its success in implementing the project in this study, it suggests that the Index for Inclusion framework should be further implemented and studied to support the implementation of strengths-based approaches in schools.

This review also reveals that teachers' attitude and knowledge are important elements of strengths-based practice. This relates to a teachers' willingness and ability to identify strengths and interests and "deliberate application of strengths within and outside of the classroom" [16]. Although inclusion of all students in education has been supported by a growing number of educational systems around the world, inclusion in schools requires teachers and schools to develop their own processes to promote strengths-based practice for students with additional learning needs, including autistic students [67]. Therefore, providing resources and professional development opportunities in developing the beliefs of teachers and building their competencies should be a priority in practice and research in the future. Despite the limited evidence due to five studies being practitioner-based articles, this review supports the recommendations made by Kasari and Smith [55]. These recommendations include to examine school-based projects and involve teachers in participatory research approaches, so that more evidence-based strengths-based approaches that suit school environments and promote long-term changes in students can be implemented.

- Limitations

There are some limitations of this review that must be considered. First, the fact that we did not expand beyond "strengths-based approaches" in our search means that some strategies and practices that could be considered to leverage the strengths and interests of autistic students may not have been included in this review. However, the search would be far too broad if we had expanded the scope. Second, only literature written in English was included, of which the majority was from the US. The lack of inclusion of non-English publications and limited representation of countries will contribute to the limitation of the perspectives and experiences of strengths-based practice. Third, the heavy skew of male autistic students in the selected studies may have limited the holistic understanding of strengths-based approaches that benefit all autistic students.

## 5. Conclusions

This scoping review demonstrated that strengths-based practice can be implemented to work with autistic students in school environments for improving both their educational experiences and outcomes. Elements contributing to strengths-based approaches for autistic students are interrelated across the different layers of the school environment. This review highlights the importance of leveraging strengths, developing relationships and adjusting environmental factors for the success of autistic students in schools. Findings of this review may assist educators and researchers to employ strengths-based approaches to support the inclusion of autistic youth in mainstream education.

**Author Contributions:** Conceptualization, J.W., S.M., M.F. and M.H.B.; methodology and analysis, J.W., S.M., M.F., M.S. and M.H.B.; writing—original draft preparation, J.W.; writing—review and editing, J.W., S.M., M.F., M.S., P.J.W. and M.H.B. All authors have read and agreed to the published version of the manuscript.

**Funding:** The authors acknowledge the financial support from the Cooperative Research Centre for Living with Autism (Autism CRC), established and supported under the Australian Government's Cooperative Research Centres Programme. The authors also wish to acknowledge the financial support of Curtin University to Jia White through the Australian Postgraduate Award Scholarship.

**Data Availability Statement:** Not applicable.

**Conflicts of Interest:** The authors declare no conflict of interest. The funders had no role in the design of the study; in the collection, analyses, or interpretation of data; in the writing of the manuscript; or in the decision to publish the results.

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
