# Peer review of "Creating Inclusive Schools for Autistic Students: A Scoping Review on Elements Contributing to Strengths-Based Approaches"

_education, doi:10.3390/educsci13070709_

Round 1

Reviewer 1 Report

This collaborative paper is very good and interesting. The authors conducted an appropriate search for work done on education strategies for children and teenagers with autism to find best practices for gaining successful outcomes by teachers for their students with autism. The focus in strength-based approaches to education is sound and informative fort educators, parents, and anyone interested in knowing how to engage people living on the spectrum.  

Minor grammatical errors. I am uncertain of and unfamiliar with the footnote style guide in use (e.g. CMS uses first name first in foot notes).

Lines 28-29, the word 'programs' seems necessary to complete the sentence "... attending schools are educational programs" [4].

Author Response

Point 1: This collaborative paper is very good and interesting. The authors conducted an appropriate search for work done on education strategies for children and teenagers with autism to find best practices for gaining successful outcomes by teachers for their students with autism. The focus in strength-based approaches to education is sound and informative fort educators, parents, and anyone interested in knowing how to engage people living on the spectrum.  

Response 1: Thank you for your positive feedback and for your comments.

Point 2: Minor grammatical errors. I am uncertain of and unfamiliar with the footnote style guide in use (e.g. CMS uses first name first in foot notes).

Response 2: The referencing format has been checked according to the instructions in the template provided by the journal. 

Point 3: Lines 28-29, the word 'programs' seems necessary to complete the sentence "... attending schools are educational programs" [4].

Response 3: The missing word ‘programs’ has been added in the sentence in Line 29.

Reviewer 2 Report

Dear author,

 You address a relevant topic, namely the educational inclusion of students with autism spectrum disorder. Further research is needed to promote the presence, participation, and achievement of this group in various societal domains.

Your work is based on "strength-based approaches" as a means to identify and leverage the strengths of students with ASD, which aligns with the fundamental principles of inclusive education. In this context, you approach the subject from Bronfenbrenner's Systems Theory. However, it would be beneficial to establish a clearer connection between the educational field and the different areas (microsystem, mesosystem, and exosystem) you discuss. I recommend relating these aspects to the "Index for Inclusion."

The "Index for Inclusion" is a framework developed by Tony Booth and Mel Ainscow for evaluating and promoting inclusive practices in schools and educational settings. It provides a tool to assess the inclusivity of an institution and guide its improvement process. The "Index for Inclusion" aligns with Bronfenbrenner's ecological systems theory by considering the microsystem, mesosystem, and exosystem within the educational context. It promotes inclusive practices by evaluating and addressing the interactions and connections within the school (microsystem), between the school and the broader community (mesosystem), and the influence of broader policies and societal norms (exosystem). Its objective is to create inclusive learning environments that cater to the diverse needs of all students.

Furthermore, in sections 3.6.2 and 3.6.3, you refer to "Elements within the microsystem" and "Elements within the mesosystem," emphasizing different curricular strategies and resources. You also conclude by discussing the benefits that these strategies offer to all students. I recommend establishing a connection between these elements and approaches related to Universal Design applied to education, such as Universal Design for Learning (UDL). Strengths-based approaches and UDL share a common focus on promoting inclusivity and maximizing the potential of all learners. They are closely related in their shared goals of inclusivity, personalization, and holistic development.

Lastly, the search period for your study covers the years from 1994 to 2021. Although it is a substantial time frame, it is important to consider the present as well. There is an almost two-year gap that is not covered, which may result in the research findings not being up-to-date enough.

Author Response

Point 1: You address a relevant topic, namely the educational inclusion of students with autism spectrum disorder. Further research is needed to promote the presence, participation, and achievement of this group in various societal domains. Your work is based on "strength-based approaches" as a means to identify and leverage the strengths of students with ASD, which aligns with the fundamental principles of inclusive education. In this context, you approach the subject from Bronfenbrenner's Systems Theory. However, it would be beneficial to establish a clearer connection between the educational field and the different areas (microsystem, mesosystem, and exosystem) you discuss. I recommend relating these aspects to the "Index for Inclusion."

The "Index for Inclusion" is a framework developed by Tony Booth and Mel Ainscow for evaluating and promoting inclusive practices in schools and educational settings. It provides a tool to assess the inclusivity of an institution and guide its improvement process. The "Index for Inclusion" aligns with Bronfenbrenner's ecological systems theory by considering the microsystem, mesosystem, and exosystem within the educational context. It promotes inclusive practices by evaluating and addressing the interactions and connections within the school (microsystem), between the school and the broader community (mesosystem), and the influence of broader policies and societal norms (exosystem). Its objective is to create inclusive learning environments that cater to the diverse needs of all students.

Response 1:  Thank you for this suggestion. We have now included discussion on the “Index for Inclusion” framework providing another tool for connecting the elements in the different layers of the systems has been added in Section 4, Lines 528-535:

“To further future research and practice into connecting all these elements across the different layers of the ecological system, the Index for Inclusion [66] can be utilised as a tool for promoting inclusion in education by considering all three dimensions, cultures, policies and practices, of a school. Only one study [47] included in this review aligned the intervention with the Index for Inclusion, but with its success in implementing the project in this study, it suggests that the Index for Inclusion framework should be further implemented and studied to support the implementation of strengths-based approaches in schools.”

Point 2: Furthermore, in sections 3.6.2 and 3.6.3, you refer to "Elements within the microsystem" and "Elements within the mesosystem," emphasizing different curricular strategies and resources. You also conclude by discussing the benefits that these strategies offer to all students. I recommend establishing a connection between these elements and approaches related to Universal Design applied to education, such as Universal Design for Learning (UDL). Strengths-based approaches and UDL share a common focus on promoting inclusivity and maximizing the potential of all learners. They are closely related in their shared goals of inclusivity, personalization, and holistic development.

Response 2: Thank you also for this suggestion. Discussion on how UDL can support strengths-based approaches has been added in Section 4, Lines 501-507:

“This indicates that to provide rigorous learning opportunities for autistic adolescents, more research and practices need to be explored in this area. For example, future research and practice may benefit from applying approaches related to Universal Design for Learning [62], by considering the WHY of learning, the WHAT of learning and the HOW of learning of each student to design learning based on their strengths and interests to achieve inclusivity and holistic development.”

Point 3: Lastly, the search period for your study covers the years from 1994 to 2021. Although it is a substantial time frame, it is important to consider the present as well. There is an almost two-year gap that is not covered, which may result in the research findings not being up-to-date enough.

Response 3: the search has been updated to cover up to June 2023. Two more studies have been identified as meeting the inclusion criteria and have been added. Section 2.2 search strategy, the results section (section 3.1, 3.2, 3.3. 3.4, 3.5) have been updated to include the two studies. All the tables and figures have been updated to reflect the two additional studies.

Reviewer 3 Report

The aim of the manuscript was to employ the Bioecological Model of Development as a framework to explore literature relating to strengths-based approaches for autistic students in schools, and to identify key elements contributing to the design and implementation of strengths-based approaches for autistic students in inclusive settings.  The introduction clearly justifies the need for such an analysis based on a model consistent with the new approach to disability and the resulting need to change the approach to the process of providing support to a student with autism in an inclusive school setting.  An appropriate research procedure was used-a scoping review process followed based on the framework proposed by Arksey and O'Malley, the application of which is comprehensively described.  Appropriate eligibility criteria were adopted. To ensure the methodological quality of the analyzed research reports, four independent reviewers were involved in compliance with the Standard Quality Assessment Criteria. The analysis of the results of the literature review was presented in an orderly and insightful manner, allowing the evaluation of previous research reports published in peer reviewed articles mainly from American journals. Through the analysis, appropriate variables were identified matched to the various subsystems of the model, allowing the field of further research on the effects of implementation of strengths-based approaches for autistic students in inclusive settings to emerge. The literature is exhaustive and representative of the issues discussed.  

Author Response

Dear Reviewer, Thank you for your positive feedback.